# Characterisation of an Isogenic Model of Cisplatin Resistance in Oesophageal Adenocarcinoma Cells

**DOI:** 10.3390/ph12010033

**Published:** 2019-02-20

**Authors:** Amy M. Buckley, Becky AS. Bibby, Margaret R. Dunne, Susan A. Kennedy, Maria B. Davern, Breandán N. Kennedy, Stephen G. Maher, Jacintha O’Sullivan

**Affiliations:** 1Department of Surgery, Trinity Translational Medicine Institute, St. James’s Hospital, Trinity College Dublin, Dublin 8, Ireland; bucklea6@tcd.ie (A.M.B.); margaret.a.dunne@gmail.com (M.R.D.); KENNES21@tcd.ie (S.A.K.); DAVERNMA@tcd.ie (M.B.D.); MAHERST@tcd.ie (S.G.M.); 2Translational Radiobiology Group, Division of Cancer Sciences, University of Manchester, Manchester Academic Health Science Centre, Christie Hospital, Manchester M20 4BX, UK; becky.bibby@manchester.ac.uk; 3UCD Conway Institute & UCD School of Biomolecular and Biomedical Science, University College Dublin, Dublin 4, Ireland; brendan.kennedy@ucd.ie

**Keywords:** oesophageal cancer, treatment resistance, cisplatin, metabolism, inflammation, radiation

## Abstract

Cisplatin (cis-diamminedichloroplatinum) is widely used for the treatment of solid malignancies; however, the development of chemoresistance hinders the success of this chemotherapeutic in the clinic. This study provides novel insights into the molecular and phenotypic changes in an isogenic oesophageal adenocarcinoma (OAC) model of acquired cisplatin resistance. Key differences that could be targeted to overcome cisplatin resistance are highlighted. We characterise the differences in treatment sensitivity, gene expression, inflammatory protein secretions, and metabolic rate in an isogenic cell culture model of acquired cisplatin resistance in OAC. Cisplatin-resistant cells (OE33 Cis R) were significantly more sensitive to other cytotoxic modalities, such as 2 Gy radiation (*p* = 0.0055) and 5-fluorouracil (5-FU) (*p* = 0.0032) treatment than parental cisplatin-sensitive cells (OE33 Cis P). Gene expression profiling identified differences at the gene level between cisplatin-sensitive and cisplatin-resistant cells, uncovering 692 genes that were significantly altered between OE33 Cis R cells and OE33 Cis P cells. OAC is an inflammatory-driven cancer, and inflammatory secretome profiling identified 18 proteins secreted at significantly altered levels in OE33 Cis R cells compared to OE33 Cis P cells. IL-7 was the only cytokine to be secreted at a significantly higher levels from OE33 Cis R cells compared to OE33 Cis P cells. Additionally, we profiled the metabolic phenotype of OE33 Cis P and OE33 Cis R cells under normoxic and hypoxic conditions. The oxygen consumption rate, as a measure of oxidative phosphorylation, is significantly higher in OE33 Cis R cells under normoxic conditions. In contrast, under hypoxic conditions of 0.5% O_2_, the oxygen consumption rate is significantly lower in OE33 Cis R cells than OE33 Cis P cells. This study provides novel insights into the molecular and phenotypic changes in an isogenic OAC model of acquired cisplatin resistance, and highlights therapeutic targets to overcome cisplatin resistance in OAC.

## 1. Introduction 

Oesophageal cancer is the sixth most common cause of cancer-related mortality globally, and unfortunately, the five-year survival rates remain low at <20% [1,2]. Oesophageal cancer consists of two different histological subtypes: oesophageal squamous cell carcinoma (OSCC) and oesophageal adenocarcinoma (OAC) [1]. In recent years, the epidemiology of oesophageal cancer dramatically shifted, wherein OAC is now the most prevalent subtype in the Western world [3]. The increasing OAC incidence in Western countries has paralleled the increasing rates of obesity, which is a known risk factor for OAC [4]. 

The current standard treatment of care for oesophageal cancer focusses on neoadjuvant treatment with chemotherapy alone (neoCT) or in combination with radiation, and neoadjuvant chemoradiation (neoCRT) for locally advanced tumours prior to surgery [5]. Despite the improvements demonstrated with neoadjuvant treatment versus surgery alone, only ~30% of patients achieve a complete pathological response to neoadjuvant treatment, which is a proxy for improved overall survival [6]. Surgery offers the best chance of locoregional control, and neoadjuvant treatment aims to reduce tumour burden prior to surgery to improve post-operative outcome. Thus, it is critical to improve response rates to neoadjuvant therapy to increase patient survival [5,7,8]. Treatment resistance is a major cause of treatment failure, and it is crucial to understand the molecular mechanisms and markers governing the treatment resistance phenotype. 

Cisplatin is a platinum-based chemotherapy that is used to treat a wide number of solid cancers [9]. Cisplatin is administered as part of a neoadjuvant chemotherapy regimen, and is referred to as MAGIC for the treatment of OAC [10]. The MAGIC chemotherapy protocol consists of the administration of epirubicin, cisplatin, and 5-fluorouracil pre-operatively and post-operatively [11,12]. One mechanism by which cisplatin exerts its anti-cancer effect is through the generation of DNA lesions, resulting in the activation of the DNA damage response and the subsequent induction of mitochondrial-mediated apoptosis [9]. Initial responses to cisplatin are often quite promising; however, chemoresistance to cisplatin frequently develops, leading to treatment failure, and a poor response to neoadjuvant treatment is associated with poor outcome. Thus, it is critical to understand the molecular mechanisms governing resistance to cisplatin in OAC. Our first findings investigate cross-resistance and sensitivity to other cytotoxic treatments, including radiation and 5-fluorouracil (5-FU). 

OAC is an inflammatory-driven upper gastrointestinal malignancy, and previous studies have reported on the role of inflammation as a negative regulator of response to radiation treatment in OAC [4,13]. Inflammation is linked with cisplatin resistance, and Cui et al. demonstrated that interleukin-7 (IL-7) upregulation was associated with cisplatin resistance in glioma cells [14]. Targeting IL-7 improved cisplatin sensitivity [14]. In the present study, we identify changes in the inflammatory secretome in an isogenic OAC model of cisplatin resistance to increase our understanding of inflammation in cisplatin resistance. Inflammation is also tightly linked to metabolism, as inflammatory cells rely on metabolites generated from the metabolic cycle to maintain their function [15]. Tumours rely on metabolism to fuel their growth, and altered energy metabolism is an emerging hallmark of cancer [16]. Metabolic alterations correlate with treatment resistance in OAC, wherein increased oxygen consumption is linked with a radioresistant phenotype in an isogenic model of OAC radioresistance [17,18]. Furthermore, in a lung cancer model, cisplatin-resistant cells have significantly elevated oxygen consumption rates compared to parental cisplatin-sensitive cells [19]. This paradigm has not yet been explored in a cisplatin-resistant model of OAC.

In this study, we examined the key differences in chemosensitivity, radiosensitivity, gene expression, inflammatory secretions, and metabolism between matched OAC cisplatin-sensitive (OE33 Cis P) cells and OAC cisplatin-resistant (OE33 Cis R) cells. Cisplatin-resistant cells are more sensitive to radiation and 5-FU treatment than cisplatin-sensitive cells. Cisplatin resistance was associated with a number of significant changes in gene expression, where KEGG pathway analysis identified four pathways upregulated in chemoresistant OE33 Cis R cells, including pathways in cancer, tumour growth factor (TGF)-beta signalling, Wnt signalling, and steroid biosynthesis. Eighteen proteins were secreted at significantly altered levels in OE33 Cis R cells compared to OE33 Cis P, with IL-7 secretion at a significantly higher level from chemoresistant OE33 Cis R cells compared with OE33 Cis P cells. Metabolic profiling revealed that the oxygen consumption rate, as a measure of oxidative phosphorylation, was significantly higher in OE33 Cis R cells under normoxic conditions (*p* = 0.0040). In contrast, under hypoxic conditions, the oxygen consumption rate was significantly lower in OE33 Cis R cells than in OE33 Cis P cells (*p* = 0.0078). This study highlights molecular and phenotypical changes in an isogenic OAC model of acquired cisplatin resistance, and highlights key differences that could be therapeutically targeted to overcome cisplatin resistance in OAC.

## 2. Results

### 2.1. OE33 Cis R Cells Are More Sensitive to Radiation and 5-Fluorouracil (5-FU) Treatment

The relative cisplatin sensitivity of the parental cell line, OE33 Cis P, and the age and passage-matched cisplatin resistant subclone, OE33 Cis R, was evaluated by clonogenic assay. The treatment of cisplatin-sensitive OAC cells with the IC_50_ of cisplatin was previously determined in CCK8 assay (Figure 1); 1.3 µM of cisplatin significantly reduced the surviving fraction of OE33 Cis P cells to 0.303 compared to untreated OE33 Cis P cells, *p* = 0.0108 (Figure 2A). However, 1.3 µM of cisplatin did not significantly alter the surviving fraction of OE33 Cis R cells (0.944 ± 0.042 compared to untreated OE33 Cis R cells), which in itself was significantly higher than the surviving fraction of the OE33 Cis P cells treated with 1.3 µM of cisplatin, *p* = 0.0011 (Figure 2A). A ~two-fold higher concentration, 2.8 µM of cisplatin, significantly reduced the surviving fraction of OE33 Cis R cells to 0.604 ± 0.045, which was a reduction of ~39%, *p* = 0.0043 (Figure 2A). Notably, OE33 Cis P cells were not clonogenically viable with 2.8 µM of cisplatin. To investigate whether OE33 cells with acquired cisplatin resistance had altered sensitivity to other treatments, we investigated the response to both clinically relevant doses of radiation and 5-FU. The basal cell survival and radiosensitivity of cisplatin-sensitive OE33 Cis P cells and cisplatin-resistant OE33 Cis R OAC cells were assessed by clonogenic assay. Basal cell survival was assessed in OE33 Cis P and OE33 Cis R to determine if in the absence of any irradiation, there was a difference in surviving fraction. No significant difference was observed between the two cell lines under basal conditions, indicating that there is no longer-term proliferation differences between these cell lines, which might correlate with the altered radiosensitivity phenotypes (Figure 2B). To assess whether acquired cisplatin resistance conferred altered radiosensitivity, OE33 Cis P and OE33 Cis R cells were either mock-irradiated or treated with a single dose of 2 Gy X-ray radiation. Interestingly, OE33 Cis R cells were significantly more radiosensitive than OE33 Cis P cells, *p* = 0.0055 (Figure 2C). Similarly, OE33 Cis R cells were significantly more sensitive to 5-FU compared to the OE33 Cis P cells following 72 h of 5-FU treatment, *p* = 0.0032 (Figure 2D). In summary, OE33 Cis R cells were more radiosensitive and 5-FU chemosensitive when compared to the parental OE33 Cis P cells.

### 2.2. Gene Expression Is Significantly Altered in OE33 Cis R Cells

OE33 Cis R cells have increased ALDH1 activity, which is a marker of stemness, compared to OE33 Cis P cells [20]. Thus, we investigated whether acquired cisplatin resistance was associated with significant changes in gene expression by whole genome digital gene expression analysis. Of the ~25,000 genes detected, 692 genes were significantly altered, 278 were upregulated, and 414 were downregulated between OE33 Cis P and OE33 Cis R cell lines. Applying a threshold of ±twofold change expression, 42 genes were upregulated (Figure 3A), and 104 genes were downregulated (Figure 3B) in the OE33 Cis R cell line compared to the OE33 Cis P cell line. Kyoto Encyclopedia of Genes and Genomes (KEGG) pathway analysis identified four pathways that were significantly upregulated in OE33 Cis R cells based on the gene expression findings, including: pathways in cancer, TGF-beta signalling, Wnt signalling, and steroid biosynthesis (Table 1A). Furthermore, eight KEGG pathways were downregulated in OE33 Cis R cells, including the MAPK signalling pathway, complement and coagulation cascades, RIG-I-like receptor signalling, sulfur metabolism, NOD-like receptor signalling, cell adhesion molecules, B cell receptor signalling, and Notch signalling (Table 1B). In summary, OE33 Cis R cells with acquired cisplatin resistance have a significantly altered gene expression profile compared to matched cisplatin sensitive OE33 Cis P cells. 

### 2.3. Acquisition of Cisplatin Resistance Results in an Alerted Inflammatory Secretome

OAC is an inflammatory-driven upper gastrointestinal cancer, and previous studies have reported inflammation as a negative regulator of response to radiation treatment in OAC [13,21]. However, a link between inflammation and cisplatin resistance in OAC has not yet been investigated. To determine if cisplatin-resistant OE33 Cis R cells have an altered inflammatory secretome to cisplatin-sensitive OE33 Cis P OAC cells, a 47-analyte multiplex enzyme linked immunosorbent assay (ELISA) was conducted. Twenty-three proteins of 47 analytes were detected in either one or both cell lines. IL-7 was secreted at significantly higher levels from OE33 Cis R cells, *p* = 0.0191 (Figure 4A). Seventeen proteins were secreted at significantly lower levels from OE33 Cis R cells compared OE33 Cis P cells, including C-reactive protein (CRP), IL-12p70, IL-10, TNFα, macrophage-derived chemokine (MDC), intracellular adhesion molecule 1 (ICAM-1), IL-6, IL-1β, IL-13, serum amyloid A (SAA), TARC, IL-4, IL-8, IL-1RA, IL-1α, IL-2, and IP-10, as shown in Figure 4 (*p* < 0.05). There was no significant difference in the secreted levels of five of the 23 proteins detected; MCP-1, VEGF-A, VCAM, GM-CSF, and IL-15 between the cell lines (Appendix A). OE33 Cis R cells have an altered inflammatory secretome compared to cisplatin-sensitive OE33 Cis P cells, where OE33 Cis R cells have reduced levels of a number of inflammatory chemokines and cytokines and vascular injury proteins except for IL-7, which is secreted at significantly higher levels from OE33 Cis R OAC cells. 

We also compared our findings from our gene expression study (Figure 3) to our inflammatory secretome data described in this section to determine if there was any overlap between the two datasets. In both our gene expression data (Figure 3B) and inflammatory secretome dataset (Figure 4G, 4N, 4P) ICAM-1, IL-8, and IL-1α were both expressed and secreted at significantly lower levels from OE33 Cis R cells compared to OE33 Cis P cells (Table 2). 

### 2.4. OE33 Cis R Cells Have an Altered Metabolic Phenotype Compared to the Parental OE33 Cis P Cells

Altered mitochondrial function is linked with treatment resistance in OAC; thus, we wanted to investigate the association between cisplatin resistance and metabolism in our model of OAC-acquired cisplatin resistance [17]. To investigate whether cisplatin-resistant OE33 Cis R cells have altered energy metabolism, we measured the two major energy pathway—oxidative phosphorylation and glycolysis—using the Seahorse XFe24 analyser under normoxic and hypoxic conditions. OE33 Cis R cells have a significantly higher (*p* = 0.0040) oxygen consumption rate (OCR), which is a measure of oxidative phosphorylation compared to OE33 Cis P cells, under normoxia. However, the oxygen consumption rate of OE33 Cis R cells was significantly lower than OE33 Cis P cells under low oxygen conditions (0.5% O_2_), *p* = 0.0078, (Figure 5A). There was no significant difference in the extracellular acidification rate (ECAR), which is a measure of glycolysis, between the two cell lines under normoxic or hypoxic conditions (Figure 5B). To further investigate changes in cellular energetics both OE33 Cis P and OE33 Cis R cells were treated with oligomycin (a mitochondrial complex V inhibitor and antimycin A (a mitochondrial complex III inhibitor), which inhibit specific processes in the electron transport chain. Oligomycin and antimycin produced a similar level of OCR inhibition in both OE33 Cis P and OE33 Cis R cells, and there was no significant difference in the rate of ATP production or proton leak between cisplatin-sensitive and cisplatin-resistant cells under normoxic or hypoxic conditions (Figure 5C,D). Furthermore, treatment of OE33 Cis P and OE33 Cis R cells with the uncoupling agent carbonyl cyanide-*p*-trifluoromethoxyphenylhydrazone (FCCP) showed no significant difference in maximal respiration rate between the cell lines (Figure 5E). In addition, the rate of non-mitochondrial respiration was not significantly altered between the two cell lines (Figure 5F). In summary, OE33 Cis R cells have an alerted metabolic phenotype whereby they have a significantly elevated OCR compared to parental OE33 Cis P cells under normoxic conditions, and OE33 Cis R cells have significantly lower OCR under hypoxic conditions. 

## 3. Discussion

Cisplatin is a widely use chemotherapeutic drug for the treatment of solid cancers, including OAC; however, the development of chemoresistance to cisplatin hinders the success of this agent [9]. Previous studies in cisplatin-resistant cancer cell lines of various tissue origin report multiple mechanisms responsible for enhanced resistance, and that the cisplatin-resistant phenotype is the result of multiple microRNA, gene, and protein expression changes [22,23,24]. Molecular mechanisms commonly associated with cisplatin resistance include alterations in DNA repair, drug influx/efflux, drug detoxification, cell cycle dysregulation, and evasion of apoptosis [25]. In this study, we sought to characterise the key differences linked to resistance in an isogenic model of OAC cisplatin resistance, including relative chemosensitivity and radiosensitivity, whole gene expression, inflammatory protein secretions, and metabolic phenotype. 

Chemoresistant OE33 Cis R cells have a significant acquired resistance to cisplatin compared to cisplatin-sensitive OE33 Cis P cells. Interestingly, cisplatin-resistant OE33 Cis R cells are more radiosensitive than matched OE33 Cis P cells. Cisplatin is often administered in combination with radiation for the treatment of numerous solid cancers, including oesophageal, head, neck, and cervical cancer [26,27,28,29]. Cisplatin has been widely shown to enhance radiosensitivity, which is potentially through the non-homologous end joining (NHEJ) DNA repair pathway [30,31]. As cisplatin is often given in combination with radiation to improve response, it was surprising that OE33 Cis R cells are more radiosensitive than OE33 Cis P cells. OE33 Cis R cells can overcome cytotoxic insult from cisplatin to a greater degree than OE33 Cis P cells, but cannot overcome radiation-induced DNA damage as efficiently as they could before the acquisition of cisplatin resistance. Whilst it was unexpected that OE33 Cis R cells are more radiosensitive, as DNA is the common target of both cisplatin and radiation, the mechanisms of action in yielding lethal events, delivery methods, and repair systems as a result of both treatments are very different. Cisplatin has to cross the plasma membrane and traffic to the nucleus; this is a process during which it can be affected in several ways, such as influx via CTR1, efflux via ABC transporters, detoxification by glutathione, trapping in lysosomes, altered trafficking between the plasma membrane and the nucleus, as well as DNA repair proficiency [32,33]. Radiation, on the other hand, largely influences DNA damage through water radiolysis (70%), and has direct effects on DNA bases (30%), but is also subject to energy transfer, glutathione scavenging, oxygen fixation, and DNA repair proficiency [34]. Radiation is responsible for the induction of double-strand breaks in DNA, whereas cisplatin treatment results in the formation of DNA adducts as a result of covalent bonds [31]. Additionally, cisplatin largely employs homologous recombination and nucleotide excision repair as its major pathways of repair, while radiation employs non-homologous end joining, base excision repair, single-strand break repair and, to a much lesser extent, homologous recombination. Thus, the DNA repair mechanics that were employed by cancer cells following either radiation or cisplatin treatment are quite different, and the significance of DNA repair in potentially driving this phenotype requires further mechanistic investigation in the future. Furthermore, OE33 Cis R cells were found to be more sensitive to 5-FU treatment compared to OE33 Cis P cells. A previous study established cisplatin-resistant and 5-FU-resistant OAC OE19 cell lines [35]. Interestingly, miRNA expression profiling in the OE19 cisplatin-resistant cell line and the OE19 5-FU-resistant cell line identified a large number of miRNA that were significantly differentially expressed in comparison to the relatively chemosensitive controls [23,31]. In the OE19 cisplatin-resistant cells 18 miRNAs were significantly dysregulated compared to controls (13 downregulated, five upregulated), of which 11 were validated including: miR-455-3p, miR-200b-3p, let-7e-5p, miR-181b-5p, miR-125a-5p, miR-181a-5p, miR-200b-5p, miR-31-5p, miR-200a-3p, miR-638, and miR-191-5p. Interestingly, the miRNA expression profile of the cisplatin-resistant cell line differed from the miRNA expression profile of the 5-FU-resistant cell line [23,31]. The differential miRNA expression profiles may be associated with the different mechanisms by which cisplatin and 5-FU induce cytotoxic damage.

The gene expression data that were presented in this study provide indications as to the potential mechanisms associated with enhanced resistance to cisplatin in the OE33 Cis R cells compared to the OE33 Cis P cells. Cell membrane transporter genes *ATP1A3*, *ABCA12*, *TMEM199*, and *TMEM22* were upregulated in the OE33 Cis R, and *TMEM156* and *TMEM42* were downregulated. The existing literature is limited with regards to these genes; however, they are members of cell membrane transporter families, some of which are associated with transport and the cellular accumulation of cisplatin, such as ATP7A/B and TMEM205 [25]. Upregulated Wnt signalling and the EMT pathway are also associated with cisplatin resistance [36,37]. The upregulation of the *PRICKLE2* gene may contribute to an increase in Wnt signalling in the OE33 Cis R cell line [38]. The *CDKN1C* gene produces the cyclin-dependent kinase inhibitor p57, which functions as a tumour suppressor in multiple cancer types [39]. Furthermore, the overexpression of p57 has been shown to enhance cisplatin sensitivity in-vitro via intrinsic mitochondrial apoptosis [40]. The downregulation of the *CDKN1C* gene in the OE33 Cis R cell line may contribute to the evasion of apoptosis and cellular resistance to cisplatin. Interestingly, TGF-β signalling, which was identified by KEGG pathway analysis to be upregulated in OE33 Cis R cells, has previously been linked to cisplatin resistance in nasopharyngeal and head and neck cancer; it would be of interest in the future to further understand the role Of TGF-β signalling in cisplatin resistance in OAC [41,42]. 

KEGG pathway analysis also revealed that the complement pathway was downregulated in cisplatin-resistant cells; this was an interesting finding, as levels of complement C3a and C4a of the classically activated pathway have previously been shown to be predictive of response to neoadjuvant chemoradiation therapy in OAC [13]. Considering this, we sought to investigate the larger inflammatory secretome of OAC cisplatin-resistant OE33 Cis R cells compared to that of cisplatin-sensitive OE33 Cis P cells. OAC is an inflammatory-driven cancer; thus, it is vital to understand the potential contributions of inflammatory secretions to cisplatin resistance. OE33 Cis R cells have a significantly altered inflammatory profile; only one protein, IL-7, was secreted at higher levels from OE33 Cis R cells. Whereas, 17 proteins (CRP, IL-12p70, IL-10, TNFα, MDC, ICAM-1, IL-6, IL-1β, IL-13, SAA, TARC, IL-4, IL-8, IL-1RA, IL-1α, IL-2, and IP-10) were secreted at significantly lower levels from in OE33 Cis R cells compared OE33 Cis P cells. Three of the proteins detected—ICAM-1, IL-1α and IL-8—were also found to be altered at the gene level, where they were both expressed and secreted at significantly lower levels from OE33 Cis R cells compared with OE33 Cis P cells. In a human glioma cancer model, the expression of IL-7 was positively correlated with the IC_50_ of cisplatin in both cell lines and glioma patient samples, and the overexpression of IL-7 further increased cisplatin resistance in glioma cancer cells [14]. This study supports our findings and further highlights IL-7 as a potential mediator of cisplatin resistance in OAC. IL-7 may be a useful therapeutic target for enhancing cisplatin sensitivity in OAC, and this warrants further investigation in the future.

Interleukin-1 alpha (IL-1α) is a potent inflammatory cytokine that plays a pivotal role in the inflammatory response [43]. IL-1α was both expressed and secreted at significantly lower levels from OE33 Cis R cells when compared to OE33 Cis P cells. Interestingly, IL-1α has previously been shown to play a key role in cisplatin sensitivity in human ovarian cancer cells, whereby IL-1α was shown to enhance sensitivity to cisplatin [44]. IL-1α given in combination with cisplatin was shown to enhance the anti-proliferative effect of cisplatin, increase cisplatin uptake, and increase DNA-platination in human ovarian OVCAR-3 cells [44]. Given that we have shown IL-1α to be expressed and secreted at a lower level in cisplatin resistant cells, it would be of interest to evaluate the potential chemosensitising effect of combining IL-1α with cisplatin in OE33 Cis R cells. 

Similarly, intracellular adhesion molecule-1 (ICAM-1) was found to be both expressed and secreted at significantly lower levels from OE33 Cis R cells compared to cisplatin-sensitive OE33 Cis P cells. The role of ICAM-1 in carcinogenesis and drug resistance is often dependent on the cancer and cell type. ICAM-1 has previously been shown to have anti-cancer activity in numerous studies through its recruitment of immune cells to the tumour [45,46,47]. Furthermore, cisplatin treatment has been shown to induce ICAM-1 expression in cancer cells; thus, it is not surprising that ICAM-1 is found at lower levels in the cisplatin-resistant cells compared to the OAC cisplatin-sensitive cells [48]. On the other hand, in an oesophageal SCC cancer study, ICAM-1 was reported to enhance cisplatin resistance and tumourigenicity in-vivo. However, it is important to note that this study was carried out in immunodeficient mice [49]. Thus, the potential of targeting ICAM-1 to alter cisplatin sensitivity in OAC needs further exploration, as it has direct effects on numerous cell types, including both cancer and endothelial cells, and is reliant on interactions with the tumour microenvironment.

Inflammation and metabolism are tightly interlinked biological processes whereby inflammatory cells rely on metabolites generated from the metabolic cycle to maintain their function. Given the significant differences in inflammatory protein secretions from OE33 Cis R cells compared to OE33 Cis P cells, we sought to investigate the metabolic profile of our model of acquired cisplatin resistance. Cisplatin-resistant OE33 Cis R cells have an altered metabolic phenotype compared to matched OE33 Cis P cisplatin-sensitive cells. OE33 Cis R cells have an ~40% higher rate of oxygen consumption rate than OAC cisplatin-sensitive cells. Interestingly, a previous study by Lynam-Lennon et al. demonstrated that increased oxygen consumption rate was linked with a radioresistant phenotype in OAC [17]. However, to date, the role of oxidative phosphorylation in cisplatin resistance in OAC has been underexplored. In a human ovarian cancer model of cisplatin resistance, SKOV3/DPP cisplatin-resistant cells had a significantly higher oxygen consumption rate when compared to SKOV3 cisplatin-sensitive cells [50]. The treatment of ovarian cisplatin-resistant cells with a Bcl-2 inhibitor was found to reduce the oxygen consumption rate and enhance cisplatin sensitivity [50]. Additionally, a study by Catanzaro et al. demonstrated that the inhibition of glucose-6-phosphate dehydrogenase could sensitise cisplatin-resistant ovarian cancer cells to cytotoxic death, further highlighting the role of the oxidative phosphorylation pathway in cisplatin resistance [51]. An elevated oxygen consumption rate has also been linked to chemoresistance with other chemotherapies, such as 5-FU. A study by Denise et al. demonstrated that 5-FU-resistant colorectal cancer cells have a significantly higher oxygen consumption rate compared to parental 5-FU sensitive cells. Furthermore, an increased oxygen consumption rate has also been linked to chemoresistance in docetaxel-resistant prostate cancer cells, where treatment with metformin to inhibit oxygen consumption was found to increase chemosensitivity [52]. Thus, it would be of interest to target oxidative phosphorylation in OAC OE33 Cis R cells as a potential mechanism to enhance cisplatin sensitivity. Metabolism is also tightly influenced by environmental conditions, whereby low oxygen conditions can result in the upregulation of hypoxia-inducible glucose transporters [53]. Thus, we investigated the metabolic rate of OE33 Cis P and Cis R cells under hypoxic conditions. The metabolic profile of OE33 Cis R cells is significantly altered under moderate hypoxic conditions (0.5% O_2_), where OE33 Cis R cells have a significantly lower oxygen consumption rate than parental OE33 Cis P cells. This result highlights the flexible nature of OE33 Cis R cells, whereby they are able to adapt to environmental conditions and reduce oxygen consumption rate in low oxygen conditions. Hypoxia plays a critical role in cisplatin resistance, and hypoxic tumours often display a high level of resistance to cisplatin [54,55]. Studies in lung cancer have reported that hypoxia can promote the activation of autophagy, resulting in chemoresistance [55]. In addition, cisplatin-resistant lung cancer cells display a hypoxia-induced upregulation of p53, resulting in the activation of p21 transcription, which results in the arrest of the cell cycle at the G0–G1 phase, ultimately reducing the effect of cisplatin [54]. As dynamic regions of hypoxia are found in most solid tumours, including OAC, it is critical to understand how OE33 Cis R cells adapt to their tumour microenvironment to determine the best method to target this cell population. The reduced OCR that is seen in OE33 Cis R under hypoxic conditions is something that needs to be further evaluated in future studies. 

This study has investigated a number of key differences in an isogenic model of OAC cisplatin resistance. Importantly, OE33 Cis R cells have a significantly altered gene expression and inflammatory secretome profile compared to cisplatin-sensitive cells. In addition, cisplatin-resistant cells have an altered metabolic profile under normal and low oxygen conditions. The molecular differences identified in this study, including the increased sensitivity to radiation and 5-FU of cisplatin-resistant cells, provides novel insight into cisplatin resistance in OAC, and has identified molecular processes that could be targeted in the future as a means to overcome cisplatin resistance and improve therapeutic outcomes for OAC. 

## 4. Materials and Methods 

### 4.1. Generation of the OE33 Cis P and OE33 Cis R Cell Lines

The human OE33 oesophageal adenocarcinoma cell line was obtained from the European collection of cell cultures. The isogenic model of cisplatin-resistant OAC; Cis P (cisplatin-sensitive) and Cis R (cisplatin-resistant) cells was generated as previously described [20]. Briefly, the original OE33 cells were split to generate two passage-matched flasks of OE33 cells. One flask was mock treated, and the other was treated with a metronomic dosing of cisplatin. Both lines were developed side-by-side and treated identically, differing only in being treated with vehicle or cisplatin until cisplatin-resistant cells were developed.

### 4.2. Preparation of Chemotherapeutic Drugs

Stock solutions of cis-diamminedichloroplatinum (cisplatin) and 5-flurouracil (5-FU) were prepared in phosphate-buffered saline (PBS) and dimethyl sulfoxide (DMSO), respectively. Solutions were sterile filtered and then aliquoted and stored at −20 °C. Prior to use, the solution was incubated at 37 °C and mixed thoroughly. 

### 4.3. Determining IC_50_ of Cisplatin for OE33 Cis P and OE33 Cis R Cells at Using Cell Counting kit-8 (CCK8) Assay

OE33 Cis P and OE33 Cis R cells were seeded at 5 × 10^3^ cells/200 µL in complete Roswell Park Memorial Institute (RPMI) 1640 medium supplemented with 10% fetal calf serum (Lonza, Basal, Switzerland) and 1% penicillin-streptomycin (Lonza, Basal, Switzerland) in a 96-well flat-bottomed plate and incubated at 37 °C, 5% CO_2_ overnight. Media was replaced with 180 µL of fresh complete RPMI and cells were treated with 20 µL of cisplatin at a range of increasing concentrations (0, 0.01, 1, 10, 50, 100, 200 µM). Cells were incubated for 48 h at 37 °C, 5% CO_2_/95% air. 10 µL of cell counting kit-8 (CCK8) assay solution (Sigma-Aldrich, Missouri, USA) was added to each well after 48 h. Cells were incubated at 37 °C, 5% CO_2_ for 1.5 to 2 h until appropriate colour development was observed. The absorbance was measured at 450 nm using a VersaMax microplate reader (Molecular Devices, Sunnyvale, CA, USA).

### 4.4. Clonogenic Assay

OE33 Cis P and OE33 Cis R cells were trypsinised, counted, and seeded at the optimised densities of 1–2.5 × 10^3^ in 1.5 mL of complete RPMI, in triplicate in six-well plates and allowed to adhere overnight. For chemosensitivity assays, cells were allowed to adhere for 24 h following which time they were treated with vehicle control or cisplatin. OE33 Cis P cells were treated with 1.3 µM of cisplatin, and OE33 Cis R cells were treated with either 1.3 µM or 2.8 µM of cisplatin. For radiosensitivity assays, cell were seeded and allowed to adhere for 48 h following which time OE33 Cis P and OE33 Cis R cells were either irradiated with 2 Gy (dose rate 1.73 Gy/min 195 KV 15 mA) or mock irradiated. Colonies were allowed to grow for 7 to 14 days, at which point they were fixed and stained with crystal violet (0.5%/ 25% *v*/*v* methanol) and allowed to air dry. Colonies consisting of 50 cells or more were counted using a colony counter (GelCount, Oxford Optronix, Oxford, UK). Plating efficiency (PE), which is the fraction of colonies formed by untreated cells, was calculated using the formula: PE = No. colonies/No. cells seeded. The surviving fraction (SF), which was the number of colonies formed following treatment, and was expressed in terms of PE, was calculated using the formula: SF = No. colonies/(No. cells seeded × PE).

### 4.5. Irradiation

Irradiation was performed using a XStrahl X-ray generator, (RS225), at a dose rate of 1.75 Gray per min.

### 4.6. RNA Extraction from Cell Lines

For total RNA extraction and purification, including miRNA, the RNeasy Mini Kit was used as per the manufacturer’s instructions (Qiagen, UK). Sterile RNase and DNase-free filter tips were used throughout all of the RNA experiments (TipOne Starlab, UK). Cell pellets (<5 × 10^4^ cells) were resuspended in 350 μl of RLT lysis buffer and pellets >five × 104 cells were resuspended in 600 µL of RLT lysis buffer. One volume of 70% ethanol was added to the lysate and mixed well by pipetting. RNeasy mini columns were inserted into the top of two-mL collection tubes. Up to 700 μL of the lysate was added to the columns, which were then centrifuged at 8000× *g* for 15 seconds at room temperature (RT). A 700-μL volume of RW1 buffer was added to the column, and centrifuged 8000× *g* for 15 seconds at RT. The flow through was discarded, and 500 μL of RPE buffer was added to the column and centrifuged at 8000× *g* for 15 s at RT. The flow through was discarded, and 500 μL of buffer RPE was added to the column and centrifuged at 8000× *g* for 2 min at RT. The flow through was subsequently discarded, and the column was transferred to a new 1.5-mL collection tube. A 30 to 50-μL volume of RNase-free water was pipetted directly onto the column membrane and centrifuged for 1 min at 8000× *g* to elute the RNA from the silica membrane of the column. The concentration and purity of the eluted RNA was measured on the NanoDrop ND-1000 (Thermo Scientific, Delaware, USA). Then, RNA extracts were stored at −20 °C in the short term (<one month) or −80 °C in the long term (>one month).

### 4.7. Digital Gene Expression Sequencing

The gene expression sequencing was outsourced to LC Sciences (Texas, USA). Briefly, six µg total RNA from three biological replicates was prepared in 50-µL of DEPC (diethylpyrocarbonate) water, five µL of 3M of NaOAc, pH 5.2 and 150 µL of absolute ethanol to give a final volume of 205 µL. Samples were stored at −80 °C prior to shipping on dry ice. High-throughput sequencing was performed using Illumina sequencing by synthesis technology. LC Sciences provided analysed gene expression data and KEGG (Kyoto Encyclopaedia of Genes and Genomes) analysis. Gene expression abundance was normalised and evaluated in FPKM (Fragments Per Kilobase of transcript per Million reads) using the Cuffdiff module of Cufflinks_v2.2.1. The q-value was representative of a false discovery rate (FDR) adjusted *p*-value < 0.05. The fold change in gene expression was calculated from the equation: log2 (OE33 Cis R FPKM/OE33 Cis P PMK). Pathway analysis was performed with EASE (Expression Analysis Systematic Explorer). The KEGG pathway *p*-value is based on the EASE score: a modified Fisher exact *p*-value that measures if the probability of the (count/list total) is more than random chance compared to the background list (pop hits/pop total), where ‘count’ is the number of significant genes in a pathway, ‘list total’ is the number of genes in the submitted list associated with the category (e.g., KEGG pathway), ‘pop hits’ is the number of genes in the background list associated with the term, and ‘pop total’ is the number of genes in the background list associated with the category. Fold enrichment was calculated as (count/list total)/(pop hits/pop total). The lower the *p*-value, the more enrichment of the term. Gene expression changes were considered significant if the *p* value was <0.05. Heatmaps were generated using the ‘pheatmap’ package for the R project for statistical computing (version 3.5.1) [56,57].

### 4.8. Multiplex Enzyme Linked Immunosorbent Assay (ELISA)

Supernatant from OE33 Cis P and OE33 Cis R cells were defrosted on ice. The secretion of cytokines and angiogenic growth factors was analysed by ELISA as per the manufacturer’s instructions. To assess angiogenic, vascular injury, inflammatory cytokine and chemokine secretions, a 47 multiplex kit spread across seven plates was used (Meso Scale Diagnostics, USA). The multiplex kit was used to quantify the secretions of CRP, Eotaxin, Eotaxin-3, GM-CSF, ICAM-1, IFN-γ, IL-10, IL-12/IL-23p40, IL-12p70, IL-13, IL-15, IL-16, IL-17A, IL-17A/F, IL-17B, IL-17C, IL-17D, IL-1RA, IL-1α, IL-1β, IL-2, IL-21, IL-22, IL-23, IL-27, IL-3, IL-31, IL-4, IL-5, IL-6, IL-7, IL-8, IL-8 (HA), IL-9, IP-10, MCP-1, MCP-4, MDC, MIP-1α, MIP-1β, MIP-3α, SAA, TARC, TNF-α, TNF-β, TSLP, VCAM-1, VEGF-A from OE33 Cis P and OE33 Cis R cell supernatants from cells that had been seeded at 250,000 cells per well for 48 h. Secretion data for all of the factors was normalised appropriately to cell lysate protein content using the BCA assay (Pierce).

### 4.9. OCR and ECAR Measurements in Cis P and Cis R Cells

OE33 Cis P and OE33 Cis R cells were seeded in five wells per treatment group at a density of 18,000 and 20,000 cells/well, respectively, in 24-well cell culture XFe24 microplates (Agilent Technologies, Santa Clara, CA, USA) at a volume of 100 μL and allowed to adhere at 37 °C in 5% CO_2_/95% air. Five hours later, an additional 150 μL/well complete cell culture RPMI medium was added. Following 48 h of incubation, media was removed and cells were washed with unbuffered Dulbecco’s Modified Eagle’s medium (DMEM) supplemented with 10 mM of glucose and 10 mM of sodium pyruvate, (pH 7.4) and incubated for one hour at 37 °C in a CO_2_-free incubator. The oxygen consumption rate (OCR) and extracellular acidification rate (ECAR) were measured using a Seahorse Biosciences XFe24 Extracellular Flux Analyser (Agilent Technologies, Santa Clara, CA, USA). Three basal measurements of OCR and ECAR were taken over 24 min consisting of three repeats of mix (three min)/wait (2 min)/measurement (3 min) to establish basal respiration. Three additional measurements were obtained following the injection of three mitochondrial inhibitors including oligomycin (Sigma Aldrich, Missouri, USA), antimycin-A (Sigma Aldrich, Missouri, USA) and an uncoupling agent Carbonyl cyanide 4-(trifluoromethoxy)phenylhydrazone (FCCP) (Sigma Aldrich, Missouri, USA). ATP turnover was calculated by subtracting the OCR post oligomycin injection from baseline OCR prior to oligomycin addition. Proton leak was calculated by subtracting OCR post antimycin-A addition from OCR post oligomycin addition. Maximal respiration was calculated by subtracting OCR post antimycin addition from OCR post FCCP addition. Non-mitochondrial respiration was determined as the OCR value post antimycin-A addition. All of the measurements were normalised to cell number using the crystal violet assay, transferring the eluted stain to a 96-well plate before reading. For seahorse experiments carried out under (0.5% O_2_) hypoxia, cells were seeded under normoxic conditions and allowed to adhere for 6 h at 37 °C 5% CO_2_/95% air, following which time they were placed at 37 °C 0.5% O_2_ 5% CO_2_ in the H35 Don whitley hypoxstation for 48 h, and seahorse measurements were carried out under hypoxia (0.5% O_2_) in the i2 Whitley station using the same protocol for measuring rate as described for normoxic conditions. 

### 4.10. Crystal Violet

Cells were fixed with 1% glutaraldehyde (Sigma-Aldrich, Missouri, USA) for 15 min at RT. The fixative was removed, and cells were washed with PBS and stained with 0.1% crystal violet for 30 min at RT. Plates were left to air dry and incubated with 50 µL of 1% Triton X-100 in PBS on a plate shaker for 30 min at RT. Absorbance was read at 595 nm on a VersaMax microplate reader (Molecular Devices, Sunnyvale, CA, USA).

### 4.11. Statistical Analysis

Statistical analysis was performed using GraphPad Prism version 5 software (GraphPad Software, CA, USA). Scientific data were expressed as mean ± standard error of the mean (SEM). SEM was calculated as the standard deviation of the original samples divided by the square root of the sample size. Specific statistical tests used are indicated in figure legends. For all of the statistical analysis, differences were considered statistically significant at *p* < 0.05. 

## Figures and Tables

**Figure 1 pharmaceuticals-12-00033-f001:**
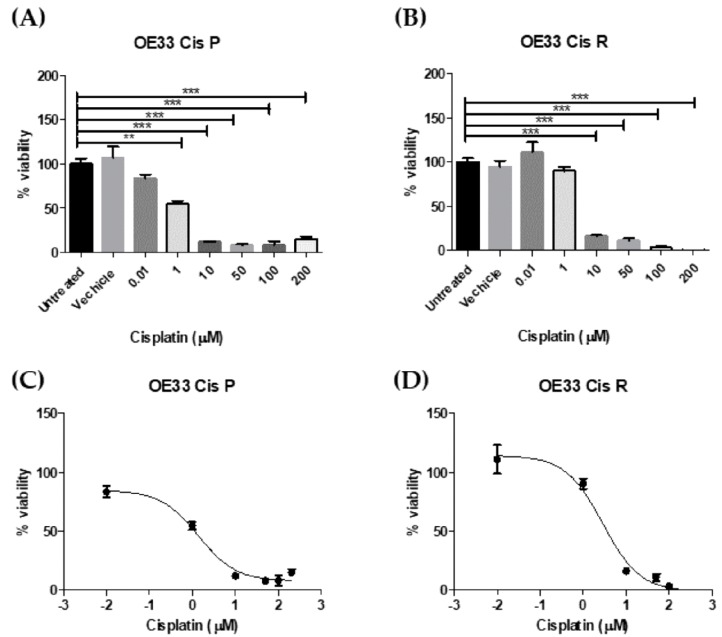
Oesophageal adenocarcinoma (OAC) cisplatin-sensitive (OE33 Cis P) cells were significantly more sensitive to cisplatin-induced cell death than OAC cisplatin-resistant (OE33 Cis R) cells. The toxicity to a range of increasing concentrations of cisplatin in (**A**) OE33 Cis P and (**B**) OE33 Cis R cells following 48 h of treatment was determined using a CCK-8 assay. The 48-h IC_50_ for (**C**) OE33 Cis P cells and (D) OE33 Cis R cells was 1.3 µM and 2.8 µM, respectively (n = 3). ** *p* < 0.01, *** *p* < 0.001 by an unpaired two-tailed *t*-test.

**Figure 2 pharmaceuticals-12-00033-f002:**
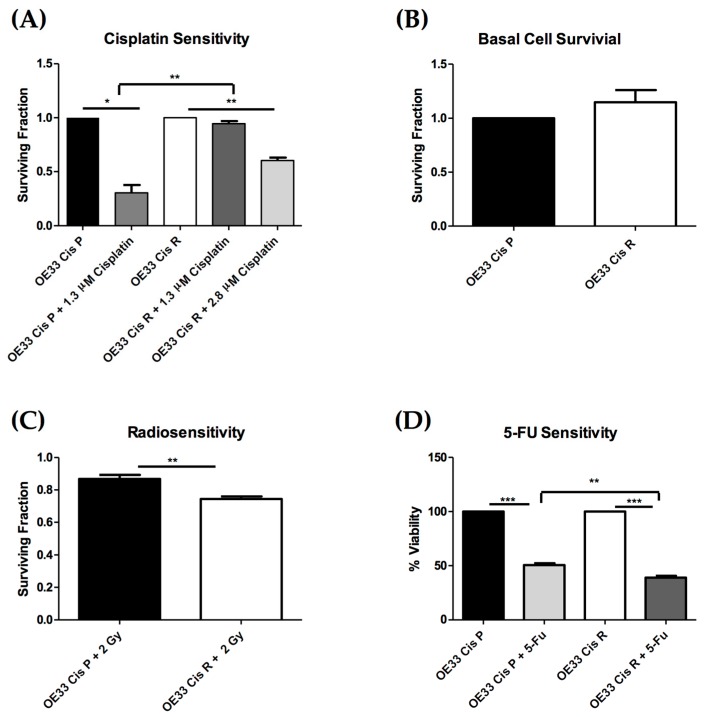
Cisplatin-resistant (OE33 Cis R) oesophageal adenocarcinoma cells are more radiosensitive than cisplatin-sensitive (OE33 Cis P) oesophageal adenocarcinoma (OAC) cells. (**A**) The sensitivity of cisplatin-sensitive (OE33 Cis P) and cisplatin-resistant (OE33 Cis R) OAC cells to cisplatin was assessed by clonogenic assay (n = 3). (**B**) There is no difference in the basal cell surviving fraction of cisplatin-sensitive and cisplatin-resistant OAC cells cultured in RPMI media, (n = 3). (**C**) Surviving fraction of Cis P and Cis R OAC cells following treatment of one 2 Gy fraction of irradiation, (n = 3). (**D**) The viability of the OE33 Cis R cells was significantly decreased compared to the OE33 Cis P when treated with 12 µM of 5-fluorouracil (5-FU) (n = 4). An unpaired *t*-test was used to compare between different cell lines, and a paired *t*-test was used to compare between the same cell line. Data presented as ±SEM * *p* < 0.05, ** *p* < 0.01, *** *p* < 0.0001.

**Figure 3 pharmaceuticals-12-00033-f003:**
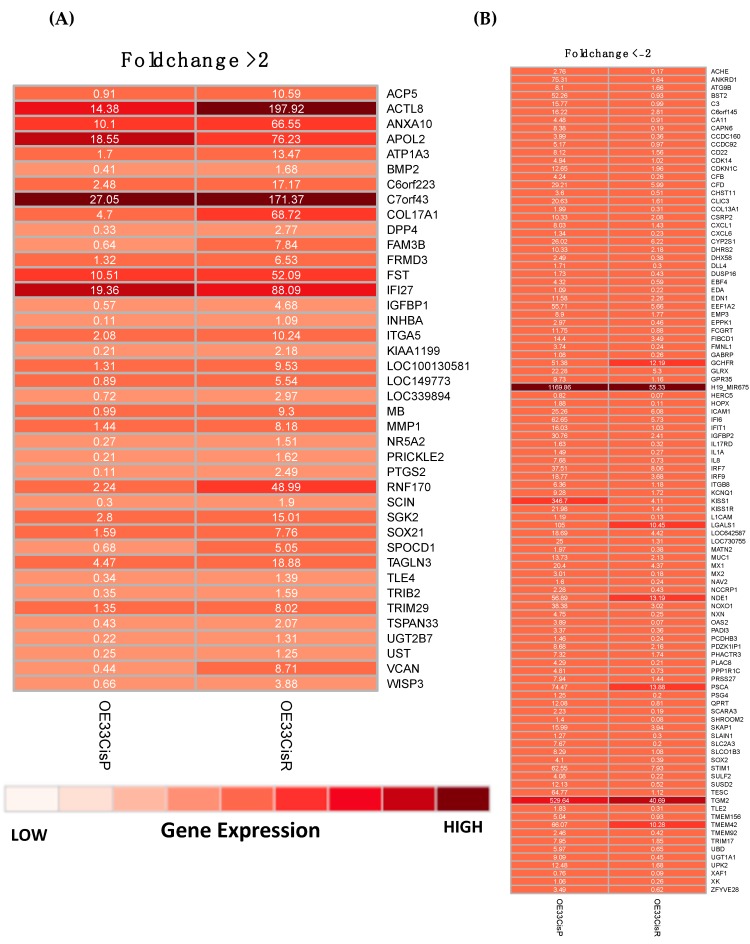
OE33 Cis R cells have a significantly altered gene expression profile compared to OE33 Cis P cells. Heatmaps were generated from gene expression data after applying a fold change filter ±two. (**A**) Heatmap showing 42 genes that were significantly upregulated in OE33 Cis R cells with a fold change of greater than two. (**B**) Heatmap showing 104 genes which were significantly downregulated in OE33 Cis R cells with a fold change of less than minus two. Gene expression values shown as Fragments Per Kilobase of transcript per Million mapped reads (FKPM).

**Figure 4 pharmaceuticals-12-00033-f004:**
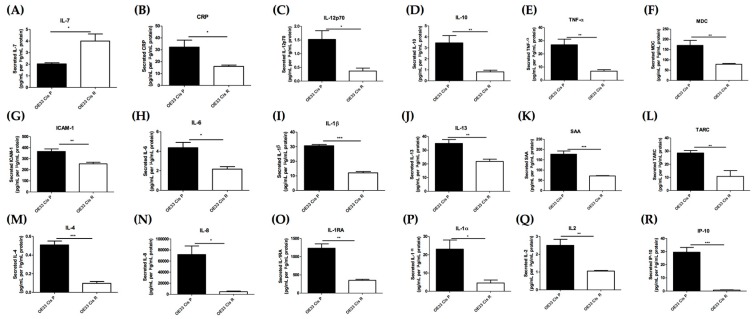
Inflammatory protein secretions are significantly different in cisplatin-sensitive (OE33 Cis P) versus cisplatin-resistant (OE33 Cis R) OAC cells. The secreted levels of 47 proteins in Cis P and Cis R cells was evaluated by multiplex ELISA; 23 proteins were detected in supernatant of Cis P and Cis R cells; 18 proteins were significantly different between the two cell lines, and interleukin-7 was significantly higher in Cis R cells compared to Cis P cells. Secreted levels of (**A**) Interleukin-7 (IL-7) (**B**) C-reactive protein (CRP) (**C**) Interleukin 12p70 (IL-12p70) (**D**) Interleukin 10 (IL-10) (**E**) Tumour necrosis factor α (TNF-α) (**F**) Macrophage-derived chemokine (MDC) (**G**) Intracellular adhesion molecule 1 (ICAM-1) (**H**) Interleukin 6 (IL-6) (**I**) Interleukin 1β (IL-1β) (**J**) Interleukin 13 (IL-13) (**K**) Serum amyloid A (SAA) (**L**) Thymus and activation regulated chemokine (TARC) (**M**) Interleukin 4 (IL-4) (**N**) Interleukin 8 (IL-8) (**O**) Interleukin 1 receptor antagonist (IL-1RA) (**P**) Interleukin 1α (IL-1α) (**Q**) Interleukin 2 (IL-2) (**R**) Interferon gamma-induced protein 10 (IP-10) in OE33 Cis P and OE33 Cis R cells, all secretions normalised to protein content. (n = 4). Unpaired *t*-test. * *p* < 0.05, ** *p* < 0.01, *** *p* < 0.001. Date expressed as ±SEM.

**Figure 5 pharmaceuticals-12-00033-f005:**
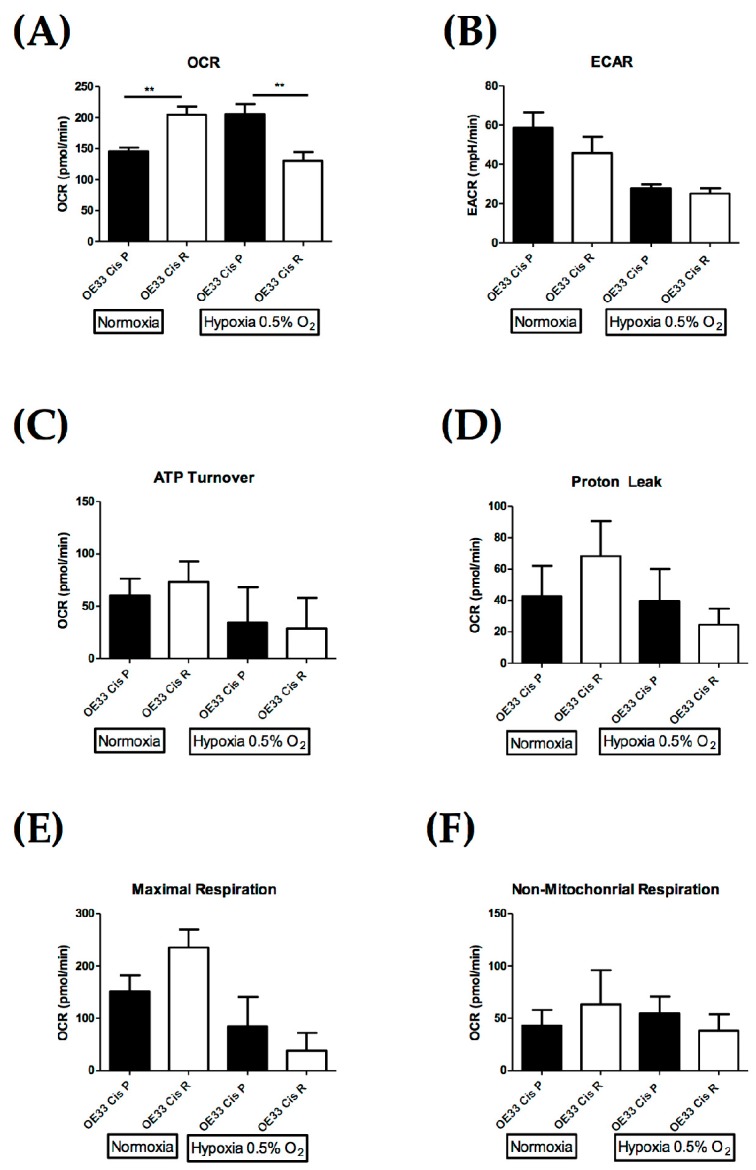
Cisplatin-resistant (OE33 Cis R) oesophageal adenocarcinoma cells have an altered metabolic phenotype compared to cisplatin-sensitive (OE33 Cis P) cells under normoxic and hypoxic conditions (0.5% O_2_). (**A**) The oxygen consumption rate (OCR), which is a measure of oxidative phosphorylation, was evaluated in OE33 Cis P and OE33 Cis R OAC cells using the Seahorse Biosciences XFe24 extracellular flux analyser cultured under normoxic and hypoxic conditions (0.5% O_2_). OE33 Cis R cells have a significantly higher OCR when compared to OE33 Cis P; cisplatin-sensitive cells, under normoxic conditions (n = 5), unpaired *t*-test, ** *p* < 0.01. (**B**) The extracellular acidification rate (ECAR), which is a measure glycolysis, was evaluated in OE33 Cis P and OE33 Cis R cells using the Seahorse Biosciences XFe24 extracellular flux analyser cultured under normoxic and hypoxic conditions (0.5% O_2_), (n = 5), unpaired *t*-test. (**C**) Difference in the rate of ATP production in OE33 Cis P and OE33 Cis R cells cultured under normoxic and hypoxic conditions (0.5% O_2_), (n = 5), unpaired *t*-test. (**D**) Difference in the rate of proton leak in OE33 Cis P and OE33 Cis R cells cultured under normoxic and hypoxic conditions (0.5% O_2_), (n = 5), unpaired *t*-test. (**E**) Difference in maximal respiration rate in OE33 Cis P and OE33 Cis R cells cultured under normoxic and hypoxic conditions (0.5% O_2_), (n = 5), unpaired *t*-test. (**F**) Difference in non-mitochondrial respiration in OE33 Cis P and OE33 Cis R cells cultured under normoxic and hypoxic conditions (0.5% O_2_), (n = 5), unpaired *t*-test. Data presented as ±SEM.

**Table molecules-19-08238-t001a:** (**A**)

KEGG Pathway Term	Number of Identified Genes Involved	*p* Value	Gene Names
hsa05200:Pathways in cancer	13	5.29 × 10^−3^	BMP4, PPARD, BMP2, BCR, PTGS2, EPAS1, PIK3CB, FOXO1, KITLG, MLH1, MMP1, RAC2, WNT9A
hsa04350:TGF-beta signalling pathway	6	1.21 × 10^−2^	BMP4, INHBA, BMP2, ID2, ID1, FST
hsa00100:Steroid biosynthesis	3	2.92 × 10^−2^	CYP27B1, LIPA, DHCR24
hsa04310:Wnt signalling pathway	6	9.19 × 10^−2^	SENP2, PPARD, RAC2, PRICKLE2, FRAT2, WNT9A

**Table molecules-19-08238-t001b:** (**B**)

KEGG Pathway Term	Number of Identified Genes Involved	*p* Value	Gene Names
hsa04010:MAPK signaling pathway	17	2.61 × 10^−4^	FGFR3, PDGFA, RELB, CACNG6, CACNG4, NR4A1, STK3, JMJD7-PLA2G4B, RASGRP3, DUSP14, JMJD7, DUSP16, RRAS, HSPB1, TRAF6, GADD45B, PLA2G4B, IL1A, DUSP6
hsa04610:Complement and coagulation cascades	7	4.23 × 10^−3^	PLAT, C3, CFB, SERPINA1, CFD, F2R, PLAUR
hsa04622:RIG-I-like receptor signaling pathway	6	2.09 × 10^−2^	IFIH1, ISG15, IL8, IRF7, TRAF6, DHX58
hsa00920:Sulfur metabolism	3	2.84 × 10^−2^	CHST11, CHST13, SULT2B1
hsa04621:NOD-like receptor signaling pathway	5	4.95 × 10^−2^	CXCL1, IL8, IL18, TRAF6, BIRC3
hsa04514:Cell adhesion molecules (CAMs)	7	7.50 × 10^−2^	ICAM1, CLDN9, CLDN3, ITGB8, PVRL2, CD22, L1CAM
hsa04662:B cell receptor signaling pathway	5	8.68 × 10^−2^	RASGRP3, CD22, PIK3AP1, MALT1, VAV1
hsa04330:Notch signaling pathway	4	8.75 × 10^−2^	HES5, DTX2, DLL4, RBPJ

**Table 2 pharmaceuticals-12-00033-t002:** Differences in gene expression and protein secretion values of ICAM-1, IL-8, and IL-1α in OE33 Cis P and OE33 Cis R cells. This table shows the mean values of data that were used to calculate the significant differences in gene expression and protein secretions previously illustrated in Figure 3B and Figure 4G, 4N, 4P. Fragments Per Kilobase of transcript per Million mapped reads (FKPM).

Protein	OE33 Cis P Mean Gene Expression (FPKM)	OE33 Cis R Mean Gene Expression (FPKM)	OE33 Cis P Mean Protein Secretion (pg/mL per µg/mL Protein)	OE33 Cis R Mean Protein Secretion (pg/mL per µg/mL Protein)
**ICAM-1**	25.26	6.08	365.64	254.39
**IL-8**	7.68	0.73	71822.10	4611.60
**IL-1^α^**	1.49	0.27	23.10	4.56

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
