# Peer review of "Characterisation of an Isogenic Model of Cisplatin Resistance in Oesophageal Adenocarcinoma Cells"

_pharmaceuticals, 2019, doi:10.3390/ph12010033_

Reviewer 1 Report

Buckley et al have developed a novel model of cisplatin resistance in oesophageal adenocarcinoma and have characterised its phenotype in relation to chemo- and radio-sensitivity, the inflammatory secretome and metabolism.  The study is well conducted and presented and the authors should be commended on the systematic approach to the development and characterisation of the cell line.  However, the study is not without its limitations and attention to a number of points would further strengthen the paper.

General Comments

The cell line model described has been developed by the repeated dosing of the parental cell line with cisplatin over a six month period.  It is unclear whether the parental cells are also grown over this six month period or are the original passage of the OE33 cells from which the resistant sub-clone is derived.  In either case, are these cell lines truly ‘isogenic’?  The molecular divergence between the parental and resistant lines which would occur over six months in culture would be considerable and so these lines should be referred to as ‘paired’ rather than ‘isogenic’.

I appreciate that the aim of the paper is to describe a broad range of phenotypes for the OE33 Cis R cells but each section needs further validation of the results. The paper reads like the start of a number of lines of enquiry, none of which are followed to completion (discussed further below).  Having identified the biology of the OE33 Cis R cells no attempt is made to modulate it or test the robustness of its association with cisplatin resistance.

Introduction

Minor Comments

Line 61- MAGIC chemotherapy consists of epirubicin, cisplatin and fluorouracil.

Line 70- role of inflammation

Line 83-85- consider re-writing as there appears to be words missing

Results

Section 2.1

Line 100- This introductory sentence is confusing.  The authors state that they are going to assess the ‘sensitivity of the cisplatin sensitive OE33’ cells.  Initially labelling the cells as ‘cisplatin sensitive’ is confusing and perhaps they should be called ‘OE33 parental cells (OE33 cis P)’

Figure 1A- the addition of images of the clonogenic plates would enhance this section of the figure.  Also, considering that a 2-fold increase from 1.3µM to 2.6 µM of cisplatin reduced the survival fraction by ~39% are these cells markedly resistant to cisplatin?  The addition of dose-response curves would enable the reader to assess the difference between the parental and resistant lines and IC30 and IC50 doses for each cell line using the CCK8 and clonogenic assays should be provided. 

Section 2.2

Was the analysis of the gene expression carried out in triplicate using three biological replicates of the cell lines?

No details are given of the software or conditions used for the pathway analysis.  Was this GSEA?  If so was the analysis corrected for multiple hypothesis testing and at what level was the FDR applied?  These factors are relevant when interpreting the results of the pathways analysis.

Figure 2A- the 42 genes shown are supposed to have been selected due to them having a >2 fold change difference between the OE33 Cis P and OE33 Cis R cells.  However, the majority of the genes are orange in colour and according to the key this indicates no change in the expression level.  A similar issue occurs with Figure 2 B.

‘OE33 Cis R cells have increased ALDH1 activity, a marker of stemness’- I cannot see the data for this statement and in the paper referenced ALDH1 activity is measured for a radio-resistant, not a cisplatin-resistant cell line.  The expression of ALDH1 should be shown in the paired cell lines and if the authors are arguing that the OE33 Cis R cells are more stem-like then other markers of stemness should also be assessed.

A major limitation of this section of the manuscript is a failure to validate any of the pathway changes outlined in Table 1.  Western blot analysis showing the protein expression of the major members of some of the pathways, MAPK, TGF-beta and Wnt signalling for example, in the paired cell lines should be included to validate the pathway changes as the potential of a false positive pathway result can be relatively high (depending upon the stringency of the pathway analysis).

Section 2.3

Again, the differences in the inflammatory protein secretions between the two paired cell lines are not examined further.  RT-PCR validation of the expression of these cytokines in the two cell lines would add to the robustness of the data.

Also there is no formal comparison of the cytokine expression detected by the multiplex ELISA and the gene expression data generated in section 2.2.  Do the same genes which are detected in higher levels by the ELISA also demonstrate upregulation of their gene expression?  There is some mention of this kind of analysis in the discussion section with the comment that ‘Three of the proteins detected, ICAM-1, IL-1α and IL-8 were also found to be altered at the gene level’.  This data should be presented in the results section and not brought into the discussion section without being mentioned previously.

Discussion

‘It was surprising that OE33 Cis R cells are more radiosensitive than OE33 Cis P cells’- This result is surprising considering that both of these agents act through DNA damage and would be expected to have similar resistance mechanisms e.g. upregulation of DNA repair proteins.  Reasons for this discrepancy should be discussed further.

The authors state ‘our data suggest the pathways that OE33 Cis R cells employ to repair DNA damage induced by either radiation or cisplatin treatment are different’ ; however no data is provide to confirm this assumption.  Of particular concern is the finding that cell membrane transport genes, such as ABCA12, are upregulated in the OE33 Cis R cells.  It is likely that the resistance to cisplatin observed in the OE33 Cis R cells is simply due to increased drug efflux and that the cisplatin does not enter the nucleus and generate DNA damage.  This could explain why the cells remain sensitive to radiation.  Further experiments should be performed to demonstrate that the cisplatin is effectively entering the cells and inducing DNA damage e.g. γH2AX foci in each cell line.

A number of sections in the discussion finish with the phrase’ needs to be further evaluated’, ‘warrants further investigation’.  This study would be greatly strengthened is some of this work was performed.  An example would be the use of siRNA/inhibitors to relevant components of the MAPK pathway in an attempt to reverse the resistant phenotype of the OE33 Cis R cells.  Similarly, the authors comment on how it would be interesting to evaluate the potential chemosensitising effect of combining IL-1α with cisplatin in OE33 Cis R cells- why not include this?

Minor Comments

Line 241- ‘and that the cisplatin resistant phenotype’

Line 245- ‘isogenic model of cisplatin resistance’

Line 290- ‘response to neoadjuvant chemoradiation therapy’

Are these cell lines being deposited in a Cell line repository so that they will be available to the wider scientific community?

Overall the authors have described the development of a novel OE33 cisplatin resistant cell line in oesophageal adenocarcinoma.  The level of resistance should be more clearly defined with dose response curves and the role of increased drug efflux should be excluded.  Whilst I appreciate that not all results can be chased up and investigated the paper would benefit greatly from validation of the pertinent results and selected investigation of key biological leads.

Author Response

Response to Reviewers Comments:

Reviewer 1

Reviewer 1, comment 1

The cell line model described has been developed by the repeated dosing of the parental cell line with cisplatin over a six month period.  It is unclear whether the parental cells are also grown over this six month period or are the original passage of the OE33 cells from which the resistant sub-clone is derived.  In either case, are these cell lines truly ‘isogenic’?  The molecular divergence between the parental and resistant lines which would occur over six months in culture would be considerable and so these lines should be referred to as ‘paired’ rather than ‘isogenic’.

Author response to reviewer 1, comment 1:

The authors thank the reviewer for their input, which has helped shape and improve the manuscript. Yes, the parental cells are age matched to the resistant sub-clone. The original passage was split, one being mock treated, the other being treated with metronomic dosing of cisplatin. Both lines were developed side-by-side and treated identically, differing only in being treated with vehicle or cisplatin. As such, this has been clarified in the paper page 13, line 444-448.

The authors did consider the appropriateness of choosing to designate the lines as ‘isogenic’ over ‘paired’, considering, as the reviewer justifiably points out, that over a six month period the two lines will have undergone potentially substantial molecular divergence, especially considering the inherent genomic unstable nature of cancer cells themselves. The authors term the model as isogenic for a number of reasons. Firstly, irrespective of the endpoint, the two lines ultimately derive from a single flask of cells, and the term isogenic generally refers to genotypes that are the same or closely similar, which was true at the beginning of the experiment. The question arises if the natural genetic drift and divergence between the parental line and the chemoresistant sub-clone over the approximate 6 months of divergence is less than, or similar to, the differences between two lines of different origin with different chemosensitivities? This is highly subjective, debatable and ultimately difficult to answer. Secondly, it was considered that the level of random genetic drift, while potentially substantial on a cell to cell basis, is unlikely to result in large numbers of statistically significant differences (genotypically and phenotypically) between the two overall cell populations, as without a selective pressure (such as cisplatin) the penetration of these random alterations is unlikely to be substantial enough to significantly alter the general population genotype and phenotype. As such, the majority of significant changes in genotype/phenotype are likely to be a direct result of changes specifically due to the selection of cells resistant to cisplatin. Thirdly, terming the paired model might imply a system involving non-isogenic cell lines at the outset, i.e. OAC cell lines with different origins and different inherent chemosensitivities. The authors feel that both ‘isogenic’ and ‘paired’ would both be appropriate to use, but that it is also context dependent, and in this case we feel that, all things considered, calling the model ‘isogenic’ is a better descriptor of the models background.  

Reviewer 1, comment 2

I appreciate that the aim of the paper is to describe a broad range of phenotypes for the OE33 Cis R cells but each section needs further validation of the results. The paper reads like the start of a number of lines of enquiry, none of which are followed to completion (discussed further below).  Having identified the biology of the OE33 Cis R cells no attempt is made to modulate it or test the robustness of its association with cisplatin resistance.

Author response to reviewer 1, comment 2:

The authors agree that there needs to be a full investigation of the biology underpinning the cisplatin resistant phenotype. However, understanding the relative contributions of significant differences between the two lines of the model require mechanistic studies, which are well beyond the scope of this initial characterisation study. The aim here was to identify the potential mechanisms involved in chemoresistance in OAC, narrowing the possibilities by using a well-controlled in-house generated model. The relative contributions of these various pathways will take considerable effort, but, as the reviewer points out, the aim here is to indeed open up our findings to the wider scientific community as it is unlikely that we will be able to pursue all lines of enquiry. 

Reviewer 1, comment 3

Line 61- MAGIC chemotherapy consists of epirubicin, cisplatin and fluorouracil.

Author response to reviewer 1, comment 3:

The authors apologise for this typo, etoposide has been corrected to epirubicin, page 2, line 61.

Reviewer 1, comment 4

Line 70- role of inflammation.

Author response to reviewer 1, comment 4:

This typo has been corrected, page 2, line 70.

Reviewer 1, comment 5

Line 83-85- consider re-writing as there appears to be words missing.

Author response to reviewer 1, comment 5:

This incomplete sentence has been corrected to read “In this study we examined the key differences in chemosensitivity, radiosensitivity, gene expression, inflammatory secretions and metabolism between matched OAC cisplatin-sensitive (OE33 Cis P) cells and OAC cisplatin-resistant (OE33 Cis R) cells.”.

Reviewer 1, comment 6           

Line 100- This introductory sentence is confusing.  The authors state that they are going to assess the ‘sensitivity of the cisplatin sensitive OE33’ cells.  Initially labelling the cells as ‘cisplatin sensitive’ is confusing and perhaps they should be called ‘OE33 parental cells (OE33 cis P)’.

Author response to reviewer 1, comment 6:

This line has been adjusted for clarity. It now reads as “The relative cisplatin sensitivity of the parental cell line, OE33 Cis P, and age and passage matched cisplatin resistant subclone, OE33 Cis R, was evaluated by clonogenic assay (Figure 1)page 3, line 102-104.The authors had assumed that the original sentence was tempered by the definition of the lines in both the abstract and introductory sections.

Reviewer 1, comment 7

Figure 1A- the addition of images of the clonogenic plates would enhance this section of the figure.  Also, considering that a 2-fold increase from 1.3µM to 2.6 µM of cisplatin reduced the survival fraction by ~39% are these cells markedly resistant to cisplatin?  The addition of dose-response curves would enable the reader to assess the difference between the parental and resistant lines and IC30 and IC50 doses for each cell line using the CCK8 and clonogenic assays should be provided. 

Author response to reviewer 1, comment 7:

The authors disagree with the reviewer’s suggestion to include images of the clonogenic plates. The clonogenic assay is frequently performed incorrectly during assessments of cellular cytotoxicity, and this is usually highlighted by the inclusion of plate images in research figures. In order to adequately power robust calculations of surviving fraction, it is necessary to optimise different cell seeding densities for different doses of chemotherapeutic agent or radiation. As the dose or drug/radiation increases the number of cells seeded into wells must also be adjusted (typically increased), so that approximately 100-300 colonies per well are obtained. The differences in seeding density are accounted for in the calculation of surviving fraction (SF), where the SF = number of colonies counted/(number of cells seeded × PE), where the plating efficiency (PE, the fraction of colonies formed in untreated wells) is calculated as the number of colonies counted/number of cells seeded. Ultimately, the result of this experimental set up is a lack of obvious dose-dependent differences in colony numbers reported across different doses, i.e. controls and treatments look similar because they are optimised to have similar numbers of resultant colonies, differing largely because of the numbers of initially seeded cells. As such the plate images will not provide an enhancement to the figures.

Reviewer 1, comment 8

Was the analysis of the gene expression carried out in triplicate using three biological replicates of the cell lines?

Author response to reviewer 1, comment 8:

The digital gene expression analysis was carried out on three biological replicates per sample. Each sample was represented by pooling of normalised RNA triplicates, i.e. 6 samples underwent DGE analysis, with each sample a pool of 3 technical replicates. This has now been clarified in the manuscript page 14, line 490.

Reviewer 1, comment 9

No details are given of the software or conditions used for the pathway analysis.  Was this GSEA?  If so was the analysis corrected for multiple hypothesis testing and at what level was the FDR applied?  These factors are relevant when interpreting the results of the pathways analysis.

Author response to reviewer 1, comment 9:

Gene expression abundance was normalised and evaluated in FPKM (fragements per kilobase of transcript per million reads using the Cuffdiff module of Cufflinks_v2.2.1. The q-value was representative of an FDR adjusted p-value<0.05. Pathway analysis was performed with EASE (Expression Analysis Systematic Explorer), as apposed to GSEA. The KEGG pathway p-value is based on EASE score, a modified Fisher exact p-value which measures if the probability of (count/list total) is more than random chance comparing to the background list (pop hits/pop total), where ‘count’ is the number of significant genes in a pathway, ‘list total’ is the number of genes in the submitted list associated with the category (e.g KEGG pathway), ‘pop hits’ is the number of genes in the background list associated with the term (e.g. MAPK signalling pathway), and ‘pop total’ is the number of genes in the background list associated with the category. Fold enrichment was calculated as (count/list total)/(pop hits/pop total). The lower the p-value the more enrichment of the term. 

Reviewer 1, comment 10

Figure 2A- the 42 genes shown are supposed to have been selected due to them having a >2 fold change difference between the OE33 Cis P and OE33 Cis R cells.  However, the majority of the genes are orange in colour and according to the key this indicates no change in the expression level.  A similar issue occurs with Figure 2 B.

Author response to reviewer 1, comment 10:

We thank the reviewer for this comment and appreciate their point about the colour scheme of this figure. These heatmaps were generated using the raw FPKM values of the OE33 Cis P and OE33 Cis R  that had a Foldchange <-2 or="">2 and was significant. Thus some values are quite low and the differences may be hard to see by colour despite the changes being significant. We have updated the colour scheme of the graph and included numerical values for better clarity, figure 2, page 6.

Reviewer 1, comment 11

‘OE33 Cis R cells have increased ALDH1 activity, a marker of stemness’- I cannot see the data for this statement and in the paper referenced ALDH1 activity is measured for a radio-resistant, not a cisplatin-resistant cell line.  The expression of ALDH1 should be shown in the paired cell lines and if the authors are arguing that the OE33 Cis R cells are more stem-like then other markers of stemness should also be assessed.

Author response to reviewer 1, comment 11:

The data demonstrating increased ALDH1 activity is not in the current article under review. This data can be found in Lynam-Lennon et al. Oncotarget, 2017;8(7):11400-11413 (Figure 3C). Panel C of figure 3 in that paper examines ALDH1 activity in the chemoresistance model, as in a counterpart radioresistance model, on which we have extensively published, ALDH1 activity was found to be associate with a radioresistance phenotype also. Figure 3C is the only data included in Lynam-Lennon et al (2017) that refers to the chemoresistance model. Taken together it is notable that ALDH1 activity is significantly higher in two independently generated OE33 resistance models, for two very different types of therapy (radiation and chemotherapy), suggesting it as an important aspect of resistance in OE33 cells. Furthermore, the DGE data demonstrates alterations in pathways well established as allied to a stemness phenotype, such as wnt and notch signalling (table 1). As this is an observational study only, we have not pursued validation of any further markers of stemness, as specific markers of stemness in OAC tumours and tumour cell lines have not yet been sufficiently defined.  

Reviewer 1, comment 12

A major limitation of this section of the manuscript is a failure to validate any of the pathway changes outlined in Table 1.  Western blot analysis showing the protein expression of the major members of some of the pathways, MAPK, TGF-beta and Wnt signalling for example, in the paired cell lines should be included to validate the pathway changes as the potential of a false positive pathway result can be relatively high (depending upon the stringency of the pathway analysis).

Author response to reviewer 1, comment 12:

This is an initial characterisation paper only, and protein-based validation of multiple pathways is outside the scope of this paper. All follow-up studies will incorporate, as the reviewer suggests, protein-based validation, allied to validation in patient samples (which we have done extensively in the past with our related radioresistance model [1]). These validation studies will ultimately involve fundamental mechanistic testing of the contributions of identified changes to the chemoresistance phenotype. Indeed, it is expected that some of the identified differences between the sensitive and resistant cells may be unrelated to the resistance phenotype, being passenger alterations as opposed to drivers. As such, validation of identified pathway changes at protein level does not answer the question of whether that pathway is a passenger or phenotypic driver within the model. Validation of individual pathways will be specific studies in their own right, again outside the scope of this paper.

Reviewer 1, comment 13

Again, the differences in the inflammatory protein secretions between the two paired cell lines are not examined further.  RT-PCR validation of the expression of these cytokines in the two cell lines would add to the robustness of the data.

Author response to reviewer 1, comment 13:

Again, the same reasoning applies here as for comment 12.

Reviewer 1, comment 14

Also there is no formal comparison of the cytokine expression detected by the multiplex ELISA and the gene expression data generated in section 2.2.  Do the same genes which are detected in higher levels by the ELISA also demonstrate upregulation of their gene expression?  There is some mention of this kind of analysis in the discussion section with the comment that ‘Three of the proteins detected, ICAM-1, IL-1α and IL-8 were also found to be altered at the gene level’.  This data should be presented in the results section and not brought into the discussion section without being mentioned previously.

Author response to reviewer 1, comment 14:

The authors appreciate this point and have clarified this in the manuscript through the inclusion of table 2, page 8. Yes, the authors compared the DGE changes against the cytokine data from the multiplex ELISA. The only proteins that were commonly changed between the two data sets were the reported ICAM-1, IL-1a and IL-8, as the reviewer points out. We have clarified this in the results section as requested page 7, line 220 -224. It is not completely surprising that there is limited overlap between the two data sets, this is frequently observed, and particularly considering one data set examines gene expression alterations, while the other examines secretion alterations. It is quite possible to have an altered cytokine secretion profile in the absence of correlating mRNA changes. However, as previously mentioned follow-up validation and mechanistic studies outside the scope of the paper will clarify this.

Reviewer 1, comment 15

‘It was surprising that OE33 Cis R cells are more radiosensitive than OE33 Cis P cells’- This result is surprising considering that both of these agents act through DNA damage and would be expected to have similar resistance mechanisms e.g. upregulation of DNA repair proteins.  Reasons for this discrepancy should be discussed further.

Author response to reviewer 1, comment 15:

This was indeed surprising. While DNA is the common target of cisplatin and radiation, the mechanisms of action in yielding lethal events, delivery method and repair systems employed by both are very different. Cisplatin has to cross the plasma membrane and traffic to the nucleus, a process during which it can be affected in several ways, such as influx via CTR1, efflux via ABC transporters, detoxification by glutathione, trapping in lysosomes, altered trafficking between the plasma membrane and the nucleus, as well as DNA repair proficiency also. Radiation, on the other hand, largely influences DNA damage through water radiolysis (70%) and direct effects on DNA bases (30%), but is also subject to energy transfer, glutathione scavenging, oxygen fixation and DNA repair proficiency. Neither of these scenarios take into account the relative burden of the stem-like population. Additionally, cisplatin largely employs homologous recombination and nucleotide excision repair as its major pathways of repair, while radiation employs non-homologous end joining, base excision repair, single strand break repair and, to a much lesser extent, homologous recombination. As such the DNA repair mechanics are quite different, and as previously mentioned the significance of DNA repair in potentially driving the phenotype needs to be clarified mechanistically. While this is outside the scope of this manuscript, we have included these points in the discussion page 10-11, line 304-324.        

Reviewer 1, comment 16

The authors state ‘our data suggest the pathways that OE33 Cis R cells employ to repair DNA damage induced by either radiation or cisplatin treatment are different’ ; however no data is provide to confirm this assumption.  Of particular concern is the finding that cell membrane transport genes, such as ABCA12, are upregulated in the OE33 Cis R cells.  It is likely that the resistance to cisplatin observed in the OE33 Cis R cells is simply due to increased drug efflux and that the cisplatin does not enter the nucleus and generate DNA damage.  This could explain why the cells remain sensitive to radiation.  Further experiments should be performed to demonstrate that the cisplatin is effectively entering the cells and inducing DNA damage e.g. γH2AX foci in each cell line.

Author response to reviewer 1, comment 16:

This statement in the discussion has been rewritten as it was unclear, page 11, line 321-324. It is already well established that the mechanisms governing repair of radiation- and chemotherapy-induced DNA damage are indeed different (please see the response to comment 15). The authors agree that the influence of ABC transporters on the chemoresistance phenotype could ideally be clarified as part of a mechanistic follow up paper. However, this raises some difficulties, for example: the gene expression data did not highlight any alterations in CTR1, the transporter largely considered responsible for influx of cisplatin, or alterations in ATP7A/B, important for cisplatin efflux. It is certainly possible that the protein levels of these transporters is altered, potentially via altered miRNA-mediated translational regulation, but this would have to be tested. It is difficult to target any of these transporters individually, as there are no specific inhibitors for doing so. Also, from experience, we know that this is a complex system to navigate – we recently published a paper in mesothelioma tumour cells where we observed increased resistance to cisplatin, which was associated with a potential increase in CTR1 levels and no change in efflux transporters. Our data demonstrated that the resistant cells had higher levels of intracellular cisplatin, but that it was sequestered in the cytoplasm. Thus, trying to determine the exact role of influx/efflux in tumour cells in relation to sensitivity in mechanistically significantly challenging. Also, the question remains that if the chemoresistance phenotype is simply a matter of altered drug flux then why are the OE33 Cis R cells more radiosensitive, considering radiation delivery does not involve membrane transporters? Again, these are all questions that need to be addressed but are well outside the scope of this manuscript.

Reviewer 1, comment 17

A number of sections in the discussion finish with the phrase’ needs to be further evaluated’, ‘warrants further investigation’.  This study would be greatly strengthened is some of this work was performed.  An example would be the use of siRNA/inhibitors to relevant components of the MAPK pathway in an attempt to reverse the resistant phenotype of the OE33 Cis R cells.  Similarly, the authors comment on how it would be interesting to evaluate the potential chemosensitising effect of combining IL-1α with cisplatin in OE33 Cis R cells- why not include this?

Author response to reviewer 1, comment 1:

Again, this is purely an observational study, and all mechanistic follow up in outside the scope of the current manuscript.

Reviewer 1, comment 18

Line 241- ‘and that the cisplatin resistant phenotype’.

Author response to reviewer 1, comment 18:

This has been changed in the revised manuscript, page 10, line 290.

Reviewer 1, comment 19

Line 245- ‘isogenic model of cisplatin resistance’

Author response to reviewer 1, comment 19.

This has been changed in the revised manuscript, page 10, line 294.

Reviewer 1, comment 20

Line 290- ‘response to neoadjuvant chemoradiation therapy’.

Author response to reviewer 1, comment 20:

This has been changed in the revised manuscript, page 11, line 357.

Reviewer 1, comment 21

Are these cell lines being deposited in a Cell line repository so that they will be available to the wider scientific community?

Author response to reviewer 1, comment 2:

No. The OE33 cell line is commercially available and we are legally prohibited from disseminating (paid or unpaid) either the parental line, or any modifications to the line, including genetic sub-clones. As such, we have detailed the procedure for generating the line, as well as including the characterisation data herein.

Reviewer 1, comment 22

Overall the authors have described the development of a novel OE33 cisplatin resistant cell line in oesophageal adenocarcinoma.  The level of resistance should be more clearly defined with dose response curves and the role of increased drug efflux should be excluded.  Whilst I appreciate that not all results can be chased up and investigated the paper would benefit greatly from validation of the pertinent results and selected investigation of key biological leads.

Author response to reviewer 1, comment 22:

As requested by the reviewer we have included the dose response curves and associated IC50 values for both the OE33 Cis P and OE33 Cis R lines, figure 1, page 4. As acknowledged by the reviewer, this is an observational study, the first paper for this model, and substantial follow-up studies will involve many of the reviewers suggested investigations, as we have previously done with our related radioresistance model.

References

1.         Lynam-Lennon, N., et al., Alterations in DNA repair efficiency are involved in the radioresistance of esophageal adenocarcinoma. Radiat Res, 2010. 174(6): p. 703-11.

Reviewer 2 Report

The article is well written and interesting, my main concern is whether the resistant cell line is a valid model.

Comments:

The IC50 value of CisPt on OE33 cells (1.3 µM) seems rather low compared to published reports (eg 4.7 µM in Due SL et al,Tamoxifen enhances the cytotoxicity of conventional chemotherapy in esophageal adenocarcinoma cells, Surg Oncol. 2016 Sep;25(3):269-77). As the IC50 value is essential to the paper, CisPt dose-response CCK8 data should be presented and at several time points (24h to 72h).

As reference 19 describes radio-resistance, generation of OE33R cells should be further described.

Assuming that Authors treated cells at IC50, the resistance index of OE33R cells is 2, a low value compared to the values of 10 or higher in most studies using resistant cell lines in cellulo. Please justify the relevance of a cell line with such a low resistance index.

Author Response

Reviewer 2

Reviewer 2, comment 1

The article is well written and interesting, my main concern is whether the resistant cell line is a valid model.

Author response to reviewer 2, comment 1:

The authors thanks the reviewer for their time and input. By its very nature this model cannot account for all aspects of tumour biology that contribute to chemoresistance, such as those dictated by microenvironmental features, such as pH, dynamic oxygen tension (hypoxia), or other cellular features of tumours, such as the stroma, vasculature, immune component. However, it is certainly useful and provides a unique opportunity to study and understand aspects of direct OAC tumour cell biology and acquired resistance that influence the chemosensitivity phenotype. It is also useful as an addition to our previously published radioresistant counterpart (Lynam_lennon et al, Radiat Res, 2010 [1]), which we have demonstrated over many years to have clinical relevance.

Reviewer 2, comment 2

The IC50 value of CisPt on OE33 cells (1.3 µM) seems rather low compared to published reports (eg 4.7 µM in Due SL et al,Tamoxifen enhances the cytotoxicity of conventional chemotherapy in esophageal adenocarcinoma cells, Surg Oncol. 2016 Sep;25(3):269-77). As the IC50 value is essential to the paper, CisPt dose-response CCK8 data should be presented and at several time points (24h to 72h).

Author response to reviewer 2, comment 2:

The authors agree that the IC50 values are low by comparison to other reports, including the one that the reviewer has mentioned. However, it should be noted that the preparation of cisplatin, vehicle (H2O, DMSO), duration of treatment, end-point assay (apoptosis, metabolic, clonogenic) and assay sensitivity all influence the IC50, and rarely are there two papers that report the same IC50 dose. Comparatively, we used a CCK8 assay to determine the IC50 values for the OE33 cells, while the paper referenced by the reviewer employed MTS assay, which has an inferior sensitivity to the CCK8 assay, thus one could expect the IC50 values to be different. As requested by the reviewer we have included the CCK8 dose-response curves from which our IC50 values were calculated, figure 1, page 4.

Reviewer 2, comment 3

As reference 19 describes radio-resistance, generation of OE33R cells should be further described.

Author response to reviewer 2, comment 3:

The generation and characterisation of the radioresistant OE33 model has been extensively described in our previous publication ‘Lynam-Lennon, Niamh, et al. "Alterations in DNA repair efficiency are involved in the radioresistance of esophageal adenocarcinoma." Radiation research 174.6a (2010): 703-711.’[1]

From Lynam-lennon et al ‘Establishment of a Radioresistant Cell Line: Cells were grown to approximately 50% confluence in vented 75- cm2 culture flasks and exposed to 2 Gy X rays (250 keV, 15 mA, 38 s) using an X-ray generator (RS225) (Gulmay Medical, Surrey, UK). Cells were trypsinized (Lonza) and subcultured once they had reached 90% confluence. This procedure was repeated until cells had received a cumulative dose of 50 Gy. Parental cells were mock-irradiated. For all assays on irradiated cells, there was at least a 10-day period between the last 2-Gy irradiation and the experiment’. This paper has now also been referenced in the manuscript and can be found as reference [18], page 2, line 80.

Reviewer 2, comment 4

Assuming that Authors treated cells at IC50, the resistance index of OE33R cells is 2, a low value compared to the values of 10 or higher in most studies using resistant cell lines in cellulo. Please justify the relevance of a cell line with such a low resistance index.

Author response to reviewer 2, comment 4:

The authors elected to use a low resistance index as it was considered that the mechanisms of chemoresistance are generally subtle. Clinically, those tumour that are ultimately treatment resistant and treatment sensitive generally do not have any major defining features, thus it is considered that the chemoresistance phenotype is largely accounted for by multiple subtle alterations in cells. Comparing cells with a higher resistance index might not be advantageous from a translational perspective. Previously, a subtle resistant index in our radioresistance model [1] revealed that the most classically described mechanisms though to govern radioresistance were not significantly different in the model, and that actual mechanisms conferring a resistant phenotype were subtle, and many of our findings validated in patient samples. We took the same approach here so the two models were comparable.

References

1.         Lynam-Lennon, N., et al., Alterations in DNA repair efficiency are involved in the radioresistance of esophageal adenocarcinoma. Radiat Res, 2010. 174(6): p. 703-11.

Round  2

Reviewer 1 Report

The authors have satisfactorily addressed the points raised in my previous review and should be commended on the reporting of a model which will be useful to the field in general.  The manuscript is significantly improved in its quality and the authors should be commended for a well presented study.

I have just one minor points:

The statistical packages and parameters used for the analysis of the gene expression data outlined in the response to my review should also be included in the manuscript.

Author Response

Reviewer 1, Revision 2, Comment 1: The statistical packages and parameters used for the analysis of the gene expression data outlined in the response to my review should also be included in the manuscript.

Author Response to Reviewer 1; Thank you for reviewing our revised manuscript, we are pleased the rebuttal satisfactorily addressed the points raised in your previous review . We have now included the parameters used for the gene expression  data out lined in the rebuttal in the manuscript pg 14, line 486-498.